



**Frequency and intensity of nitrogen addition alter soil inorganic sulfur fractions**

**but the effects vary with mowing management in a temperate steppe**

Tianpeng Li [a, 1], Heyong Liu [b, 1], Ruzhen Wang [a,*], Xiao-Tao Lü [a], Junjie Yang [c],

Yunhai Zhang [c], Peng He [a], Zhirui Wang [a], Xingguo Han [c], Yong Jiang [a,*]

[a] Institute of Applied Ecology, Chinese Academy of Sciences, Shenyang 110016,

China

[b] College of Land and Environment, Shenyang Agricultural University, Shenyang

110866, China

[c] State Key Laboratory of Vegetation and Environmental Change, Institute of Botany,

Chinese Academy of Sciences, Beijing 100093, China

[1] These authors contribute equally to this work.

[*] Correspondence to:

Dr. Ruzhen Wang, Tel: +86 24 83970603; Fax: +86 24 83970300; E-mail address:

ruzhenwang@iae.ac.cn

Dr. Yong Jiang, Tel: +86 24 83970902; Fax: +86 24 83970300; E-mail address:

jiangyong@iae.ac.cn



## Abstract

Sulfur (S) availability plays a vital role in driving functions of terrestrial ecosystems,

which can be largely affected by soil inorganic S fractions and pool size. Enhanced

ecosystem nitrogen (N) input can significantly affect soil S availability, but it still

remains largely unknown if the N effect varies with frequency of N addition and

mowing management in grasslands. To investigate changes in soil S pool and

inorganic S fractions (water-soluble S, adsorbed S, available S, and insoluble S), we

conducted a field experiment with different frequencies (twice *vs.* monthly additions

per year) and intensities (i.e. 0, 1, 2, 3, 5, 10, 15, 20, and 50 g N m$^{-2}$ year$^{-1}$) of

NH$_4$NO$_3$ addition and mowing (unmowing *vs.* mowing) over six years in a temperate

grassland of northern China. Soil water-soluble and adsorbed S concentrations

significantly increased, while insoluble S decreased with increasing intensity of N

input. Such changes were correlated with soil pH and total inorganic nitrogen (TIN)

concentration. High frequency of N addition increased the concentrations of

water-soluble S, adsorbed S and available S as compared to low frequency of N

addition in mown plots. Mowing significantly decreased all soil inorganic S fractions

by reducing S replenishment via plant residue return. Mowing significantly interacted

with both N addition intensity and frequency to affect inorganic S fractions, in that

adsorbed S and available S showed no response to N addition intensity in unmown

plots but significantly increased in mown plots under high N frequency. Mowing

interacted with N addition intensity to decrease soil S pool size, suggesting that





biomass removal under N input would cause soil S depletion in this temperate

grassland. Nitrogen addition could replenish soil available S by promoting dissolution

of soil insoluble S with decreasing soil pH and mineralization of organic S due to

increasing plant S uptake. Our results further indicated that using large and infrequent

5    N addition to simulate N deposition can overestimate the main effects of N deposition

and mowing on soil S availability in semi-arid grasslands.

**Keywords:** sulfur availability, nitrogen deposition, nitrogen input frequency, biomass

removal, soil acidification, semi-arid grassland



## 1 Introduction

Sulfur (S) is an essential nutrient for the metabolism of plants and soil microorganisms

by constituting amino acids of cysteine and methionine (Blum et al., 2013). It also

plays vital roles in increasing plant nitrogen (N) use efficiency, enhancing crop yield

and quality (De Bona and Monteiro, 2010), and reducing plant diseases and heavy

metal toxicity (Chiang et al., 2006; Feechan et al., 2005). Plant S deficiency is widely

distributed in global ecosystems (Kost et al., 2008; Scherer, 2009), with negative

effects on stomatal conductance, photosynthetic rate, and consequently on primary

productivity (Juszczuk and Ostaszewska, 2011; Wulff-Zottele et al., 2010).

As a macronutrient for plant, S occurs in soils in both organic and inorganic forms. Soil

organic S includes ester-bonded S, C-bonded S and residual S, with the former two

forms constituting the potential S source for plants. Inorganic S, accounting for

approximately 5% of total soil S in most temperate soils (Tabatabai, 2005), generally

occurs as bioavailable $SO_4^{2-}$ (including water-soluble $SO_4^{2-}$ and adsorbed $SO_4^{2-}$) and

insoluble $SO_4^{2-}$ coprecipitated with $CaCO_3$ (Fig. 1). Changes in the soil inorganic S

pool play a major role in S dynamics because it can impact the transformation of soil

organic S and plant S uptake (McGill and Cole, 1981; Ghani et al., 1993; Kertesz and

Mirleau, 2004). A deeper understanding on transport and transformation of soil

inorganic S fractions is essential for better predicting S supply to plants under global

change scenarios.

As mainly caused by increasing atmospheric N deposition and fertilization,



enhanced ecosystem N inputs result in severe ecological problems in temperate

ecosystems worldwide (Bobbink et al., 2010), which is predicted to deteriorate in the

coming decades, especially in developing countries (Dentener et al., 2006). Higher N

input may increase plant S uptake and affect soil S turnover by enhancing primary

productivity, especially in N-limited regions (De Bona and Monteiro, 2010; Harpole

et al., 2007; Phoenix et al., 2012; Wang et al., 2015). Nitrogen input can promote S

supply by stimulating the mineralization of C-bonded S (into the form of $SO_4^{2-}$-S) (De

Bona and Monteiro, 2010) and abiotic dissolution of mineral-bound S under the

N-induced soil acidification (Wang et al., 2016). However, results from an 80-year

fertilization experiment showed that N addition did not change the concentrations of

soil inorganic and organic S (Yang et al., 2007). Nitrogen inputs may also have

negative effects on S cycling rate due to the inhabitation of arylsulfatase activity

(Chen et al., 2016). Therefore, soil S availability is mainly associated with soil pH and

mineralization of soil organic matter (SOM) under N addition, but it is still poorly

understood for its relationship with inorganic S fractions.

Studies simulating N deposition commonly add N as an intensive and pulsed

input in either dry or wet form (Smith et al., 2009). However, natural N deposition

occurs more frequently and evenly in small events (Aneja et al., 2001). Low

frequency of N addition increases plant biomass (Barton et al., 2008; Bilbrough and

Caldwell, 1997) and ammonia volatilization (Zhang et al., 2014a) but decreases soil

pH (Wang et al., 2018) and plant N concentrations (Cheng et al., 2009) as compared



to high frequency. Low frequency of N addition has been reported to over-estimate

the effects of N deposition on plant species diversity (Zhang et al., 2014b). Therefore,

lower soil pH and enhanced plant biomass could possibly promote inorganic S

dissolution and plant S retention as affected by low frequency of N addition. Though

these studies demonstrated that N input frequencies alter factors associated with

transformation of soil S fractions (i.e. soil pH, plant uptake), how frequency of N

input influence transformation of soil inorganic S fractions still remains largely

unknown.

Mowing is a common management practice in temperate grasslands (Giese et al.,

2013), which greatly reduces nutrient return from plant residues (Janzen and Ellert,

1998). Persistent harvesting and mowing could decline incorporation of plant S into

soil, break the natural cycling of S, thus cause depletion of soil inorganic S (Solomon

et al., 2001). Moreover, mowing can alter soil S mineralization and immobilization by

influencing soil moisture, microbial activity and plant biomass allocation (Barrow,

1960). Under N enriched conditions, mowing would aggravate depletion of soil S

pool by removing more plant biomass and plant S out of ecosystems as compared to

ambient N condition. The effects of biomass removal on soil inorganic S fractions

remain poorly understood, while, to our knowledge, interactive N-mowing effect has

not been explored in temperate grassland ecosystems.

Temperate grasslands, which account for 8% of the earth's land surface (White et

al., 2000) with important ecological function and economic value, play an important



role in global S cycle. Primary productivity of temperate grasslands is mainly limited

by N availability and typically sensitive to N addition (Niu et al., 2010; Yang et al.,

2012). Low background N deposition in temperate grasslands of Inner Mongolia (<

1.5g N $m^{-2}$ $yr^{-1}$) makes this area an ideal place to investigate ecosystem responses to N

enrichment (Zhao et al., 2017). For better understanding soil S supply and turnover

under mowing and different intensity and frequency of N addition, a field experiment

was conducted to investigate soil inorganic S fractions and their transformations in a

temperate steppe of Inner Mongolia. We hypothesized that 1) higher intensity of N

addition would increase available S (water-soluble S and adsorbed S) concentration by

promoting insoluble S dissolution with drop of soil pH and organic S mineralization; 2)

the increase of available S and decrease of insoluble S would be more pronounced

with low frequency of N addition due to lower soil pH condition than the high

frequency; 3) mowing would decrease soil inorganic S fractions resulting from reduced

plant residue return, and such effect would be exacerbated with increasing N addition

intensities due to enhanced plant S retention.

## 2 Materials and methods

### 2.1 Site description and experimental design

The study site (43°13′ N, 116°14′ E) is a typical temperate semi-arid steppe at the

Inner Mongolia Grassland Ecosystem Research Station (IMGERS) in the Xilin River

watershed, Inner Mongolia, China. The mean annual air temperature is 0.9 °C,

varying from -21.4 °C in January to 19.7 °C in July. The long-term mean annual



precipitation is 351.4 mm, about 72.8% of which is concentrated from May to August,

according to the data from 1980 to 2013 (monitored and provided by IMGERS). The

steppe is dominated by *Leymus chinensis*, *Stipa grandis*, *Agropyron cristatum* and

*Koeleria cristata*. The soil was classified as a Haplic Calcisol by the Food and

Agriculture Organization of the United Nations (FAO) soil classification system, with a

depth of 100–150 cm and a composition of 21% clay, 60% sand, and 19% silt on

average (Hao et al., 2013). The site experienced an uncontrolled heavy sheep grazing

since 1980s and has been fenced since 1999. No fertilizers were applied before the

experiment was conducted.

The experiment was set up in 2008 following a randomized block design. There

were nine N addition intensities (0, 1, 2, 3, 5, 10, 15, 20, 50 g N m$^{-2}$ yr$^{-1}$) crossed with

two N addition frequencies (2 times yr$^{-1}$ *vs.* 12 times yr$^{-1}$) and two mowing regimes

(unmowing and mowing). $NH_4NO_3$ (> 99.5%) was added in wet and dry forms to

simulate the wet and dry N deposition, respectively. For the high-frequency

treatments, $NH_4NO_3$ was added monthly since 1st September 2008. For the

low-frequency treatments, $NH_4NO_3$ was added on the 1st of June and November since

November 2008. During the growing season (from May to October), N was added in

wet form by mixing $NH_4NO_3$ with purified water (9.0 L water in total for each plot,

either 9.0 L once in June for low frequency or 1.5 L monthly from May to October for

high frequency) and then sprayed evenly with a sprayer. To simulate dry N deposition,

$NH_4NO_3$ was mixed with treated sand to make sure even-fertilizing in non-growing



season from November to next April. Specifically, 500 g sand was used once in

November for low-frequency treatments, and 80 g sand was used monthly from

November to next April for high-frequency treatments and then broadcast evenly in

every plot. The sand used in this experiment was sieved through a 1-mm sieve, dipped

5    in hydrochloric acid, washed in purified water and then oven-dried at 120 °C for 48 h.

Annual mowing was conducted at the end of August using a hay mower at 10-cm

height to simulate the overgrazing and hay-cutting management. The aboveground

plant residue was taken away immediately after mowing. We also set an unnamed

control without any treatment (N addition, mowing, water or sand addition) to

10   determine the impacts of water and sand addition, which was also compared with

mowing treatment without any other treatments. Thus, there were 38 treatments with

10 replicate blocks for every treatment. Each plot is 8 m × 8 m and separated by a 1-m

buffer zone.

**2.2 Plant and Soil sampling**

We assessed aboveground biomass in each plot by clipping a 1 m × 1 m quadrat above

soil surface in late August 2014. All the living plants were clipped and sorted by

species, dried and weighed. The plant samples were washed using deionized water

and then dried to constant weight at 65 °C for 48 h.

Soil samples were collected from each of the 380 plots at the beginning of August

2014 (i.e. after six years of treatments). A mixed sample was taken randomly from

five cores of the topsoil (0-10cm) within each plot. Then, samples were passed





through a 2-mm sieve immediately to remove the plant residues and then air-dried for

further analysis.

### 2.3 Measurement of soil chemical properties

Soil pH was determined in a soil slurry at 2.5:1 (w/v) water: soil ratio by a digital

pH meter (Precision and Scientific Crop., Shanghai, China). The concentration of soil

organic carbon (SOC) in the topsoil was determined by oxidation using a mixture of

$K_2Cr_2O_7$ solution and sulphuric acid and titration using $FeSO_4$. Soil total inorganic

nitrogen (TIN) concentration was calculated as the sum of ammonium and nitrite,

which was extracted with 2 $M$ KCl and determined using a continuous flowing

analyzer (SANplus segmented flow analyzer, Scalar, The Netherlands). Soil pH, SOC

and TIN were previously reported in Wang et al. (2018). Soil total sulfur (TS)

concentration was analyzed with an elemental analyzer (Vario MACRO cube,

Elementar Analysensysteme GmbH, Germany).

All the S fractions were extracted at 5:1 (w/v) water: soil ratio and quantified by

turbidimetry with 0.5g of $BaCl_2$ crystals at 440nm using a UV-VIS spectrophotometer

(UV-1700, Shimadzu, Japan) (Tabatabai and Bremner, 1972). Briefly, 5.0 g of the

air-dried soil was mixed with 25ml extractant ($0.01M$ $Ca(H_2PO_4)_2$ for available S,

0.01 $M$ $CaCl_2$ for water-soluble S and 1 $M$ HCl for insoluble S) (Roberts and Bettany,

1985) and shaken at 400 rpm for 60 min at 25 °C. The extracts were then filtered and

digested with 1ml of $H_2O_2$ for 20 min to decompose organic matter. The cooled

solutions were mixed with 0.5ml of HCl (1:4) and 1 ml of acacia solution successively



and adjusted to 25 ml. The concentration of sulfur in the mixture was determined by

turbidimetry. Adsorbed sulfur, total inorganic sulfur and organic sulfur concentrations

were calculated as follows: adsorbed S = available S – water-soluble S, total inorganic

S = available S + insoluble S, and organic S = total S – total inorganic S.

We measured total S concentration in two dominant species of *Leymus chinensis*

and *Stipa grandis* in N addition plots of 0 and 15 g N m$^{-2}$ year$^{-1}$ under low and high

frequency of N addition with and without mowing. Briefly, 0.3 g plant samples were

acid digested with a 1:2 (v/v) mixture of 65% nitric acid and 72% perchloric acid

around 235°C. The S concentration of digestion solution was quantified by

turbidimetry at 440 nm using a UV-VIS spectrophotometer (UV-1700, Shimadzu,

Tokyo, Japan). Plant S uptake of dominant species was calculated as follow:

$$S_{uptake} = \sum (S_i \times m_i) \ ,$$

where $S_i$ represents S concentration in plants and $m_i$ represents aboveground biomass

of the corresponding species.

**2.4 Statistical analysis**

The Kolmogorov-Smirnov test and Levene's test were executed to determine the

normality of data and homogeneity of variances, respectively. The TIN was

$log_{10}$-transformed for homogeneity. We used three-way ANOVAs to determine the

effects of N addition intensity, N addition frequency, mowing, and their interactions

on the concentrations of soil inorganic S fractions and total S. Student's t-test was

performed to estimate the difference in plant biomass and S uptake between two N



frequencies within each N addition intensity with or without mowing. Correlation

analyses were conducted to estimate the relationships between soil parameters and

concentration of soil S fractions. The proportion of each S fraction in total inorganic S

was calculated and Duncan's HSD post-hoc test was employed to estimate differences

among treatments. All the analyses above were conducted using SPSS 18.0 (SPSS Inc.,

Chicago, IL, USA) and all statistical significance was accepted at $P < 0.05$. Moreover,

we calculated the response ratio of available S concentration to mowing practice as

follow:

$$\text{Response ratio} = \frac{S_{mown}}{S_{unmown}} \ ,$$

where $S_{mown}$ and $S_{unmown}$ represent available S concentration in mown and unmown

plots, respectively. Weighted log response ratio ($log_e$RR) and its 95% confidence

intervals for the effect of mowing were calculated using the metafor package in R

software, ver. 3.5.1. Confidence intervals not overlapping zero indicated significant

mowing effects on available S concentration.

Structural equation modeling (SEM) was conducted to examine the direct and

indirect strength of N addition intensities and frequencies on soil inorganic S fractions

through the changes in soil parameters with the AMOS 24.0 (Amos Development Co.,

Greene, Maine, USA). Data were fitted to the model using the maximum likelihood

estimation method. We used $\chi^2$-test ($P > 0.05$), root square mean errors of

approximation (RMSEA, < 0.08), and Akaike Information Criteria (AIC) to evaluate

the adequacy of the model.



## 3 Results

### 3.1 Effects of N addition and mowing on soil characters

Soil pH significantly decreased and TIN increased along the N gradient under all treatments of addition frequency and mowing regime (Table 1). Soil pH exhibited a

sharper decrease in low N frequency as compared to high N frequency in unmown plots, especially at N addition intensity of 10 and 15 g N m$^{-2}$ year$^{-1}$. Low frequency of N addition increased TIN in both unmown and mown plots at high N levels ($P < 0.05$ at 50g N m$^{-2}$ year$^{-1}$). SOC decreased along the N gradient only under low N frequency in mown plots.

### 3.2 Effects of N addition and mowing on soil inorganic S fractions

*Soil water-soluble S*

Mowing significantly decreased soil water-soluble S concentration by up to 47% at the two N frequencies (Fig. 2a, b; Table 2). High frequency of N addition significantly increased water-soluble S concentration by up to 90% with both unmowing and

mowing practices, resulting in a significant M×F effect (Fig. 2a, b; Table 2). At high frequency of N addition, intensity of N addition increased water-soluble S for both unmown (Fig. 2a, *P*=0.004) and mown plots (Fig. 1b, *P*=0.001), causing a significant F×N effect (Table 2).

*Soil adsorbed S*

Soil adsorbed S concentration was significantly affected by mowing treatments and intensity of N addition (Table 2). Mowing significantly decreased soil adsorbed S



concentration at low N addition frequency (Fig. 2c, d; Table 2). There was no effect of

N addition frequency on adsorbed S (Table 2). As compared to control plot, intensity

of N addition increased soil adsorbed S concentration at low frequency of N addition

in unmown plots (Fig. 2c, $P < 0.01$) and at both low and high frequency of N addition

(Fig. 2d, $P$=0.04 and 0.01, respectively) in mown plots, causing significant M×F and

F×N effects (Table 2).

*Soil insoluble S*

Mowing decreased soil insoluble S concentration by 55% irrespective of both

intensity and frequency of N addition (Fig.2 e, f). There was no significant effect of N

addition frequency on soil insoluble S (Table 2). Intensity of N addition decreased soil

insoluble S concentration at both low and high N frequency for unmown plots (Fig.2e;

Table 2, $P = 0.02$ and <0.01, respectively), but only at high N addition frequency for

mown plots (Fig.2f, $P < 0.01$), resulting in significant M×F and M×N effects on soil

insoluble S concentration (Table 2).

**3.3 Effects of N addition and mowing on soil available S**

Mowing significantly decreased soil available S concentration by up to 43% and 40%

at low and high N addition frequency, respectively (Fig. 3a, b; Table 2). For mown

plots, high frequency of N addition significantly increased soil available S

concentration by up to 57% (Fig. 3b; Table 2). High intensity of N addition increased

soil available S concentration at lower frequency of N addition for unmown plots

from 15.9 to 24.0 mg kg soil$^{-1}$ (Fig. 3a, $P < 0.01$), and at both low (from 10.7 to 15.6



mg kg soil$^{-1}$, $P = 0.01$) and high (from 12.0 to 23.0 mg kg soil$^{-1}$, $P < 0.01$) frequencies

of N addition for mown plots (Fig. 3b). This resulted in significant interactive effects

of M×F, F×N, and M×F×N on soil available S (Table 2).

Nitrogen addition increased the negative effect of mowing on soil available S

concentration at low frequency of N addition (Fig. 3c, $P < 0.01$). However, negative

mowing effect on available S concentration decreased along the increasing N addition

intensity and even turned into positive at 15, 20 and 50 g N m$^{-2}$ year$^{-1}$ at high

frequency of N addition (Fig. 3c, $P < 0.01$). Proportion of available S fraction (sum of

water-soluble and adsorbed S) responded differently to N addition and mowing (Fig.

4). For unmown plots, high intensity of N addition enhanced the proportion of

available S at low frequency of N addition (Fig. 4a, from 43.2 to 64.7%), while it

showed no impact at high frequency of N addition (Fig. 4b). In contrast, soil available

S proportion increased with increasing N addition intensity at high instead of low

frequency of N addition for mown plots (Fig. 4c *vs*. 4d).

**3.4 Effects of N addition and mowing on soil total inorganic S, total S, plant**

**biomass and plant S uptake**

Effects of mowing, frequency of N addition, and intensity of N addition were

significant on soil total inorganic S (Table 2). Mowing significantly decreased total

inorganic S at both low and high frequency of N addition by up to 53% (Fig. 5a, b). In

mown plots, high frequency of N addition significantly increased soil total inorganic S

concentration by as much as 30% comparing to low N frequency (Fig. 5b; Table 2).



Nitrogen addition increased soil total inorganic S along the N gradient at both low

(from 19.7 to 25.4 mg kg soil$^{-1}$, $P$=0.01) and high N addition frequency (from 24.0 to

31.2 mg kg soil$^{-1}$) of mown plots (Fig. 5b). Soil total S concentration was not affected

by intensity of N addition, frequency of N addition, or mowing treatments (Table 2).

However, the mean value of soil total S concentration tended to decrease with

increasing N addition intensity as suggested by its linear correlation with N addition

intensity at both low ($P = 0.04$) and high ($P = 0.03$) N addition frequency in mown

plots (Fig. 5d).

High intensity of nitrogen addition increased aboveground biomass regardless of

N addition frequencies and mowing management (Fig. S1). Low frequency of N

addition increased aboveground biomass at 2 and 15 g N m$^{-2}$ year$^{-1}$ as compared to

high frequency of N addition in unmown plots. Sulfur uptake of *Leymus chinensis* and

*Stipa grandis* showed no response to mowing management in plots without N addition,

but it increased with mowing under both frequencies of N addition. Low frequency of

N addition increased S uptake of dominant species at 15 g N m$^{-2}$ year$^{-1}$ in both mown

and unmown plots (Fig. 6). Nitrogen addition intensity increased plant S

concentration of two dominant species (Fig. S2) and relative biomass proportion of

*Stipa grandis* (Fig. S3).

**3.5 Relationships between soil characters and S fractions**

Soil pH was negatively correlated with adsorbed S, available S and total inorganic S

concentrations under both low (Fig. 7a) and high frequency of N addition (Fig. 7b)



across intensity of N addition and mowing treatments (all $P < 0.01$). However,

insoluble S was positively correlated with pH only under low frequency of N addition

($P<0.01$, Fig. 7a). Soil TIN was positively correlated with adsorbed S and available S

under both low and high frequency of N addition ($P < 0.05$, Fig. 7a, b). Soil organic

carbon was positively correlated with water-soluble S only in high N frequency plots.

Under high N frequency, water-soluble S was negatively correlated with adsorbed S

and insoluble S concentrations, and adsorbed S was positively correlated with

insoluble S concentration ($P < 0.01$). Organic S concentration was negatively

correlated with adsorbed S, available S and total inorganic S concentrations at high

frequency of N addition (Fig. 7b).

Results of SEM showed that both N addition intensity and mowing practice had

direct and indirect effects on soil S fractions under both N frequencies (Fig. 7c, d).

Nitrogen addition intensity had a significantly direct and negative effect (standardized

effect size: -0.17) on insoluble S at low N frequency (Fig. 7c), and it was an indirect

effect (standardized effect size: -0.26) by altering TIN at high N frequency (Fig. 7d).

Nitrogen addition intensity only had indirectly positive effect on adsorbed S

(standardized effect size: 0.47) at low N addition frequency, while it showed both

direct and indirect effects at high N frequency (Fig. 7c, d). Nitrogen addition intensity

had indirect and negative effect on organic S concentration by altering pH at both N

addition frequencies. Mowing had directly negative effects on soil insoluble S,

adsorbed S and water-soluble S concentrations at both N frequencies, with stronger



effect sizes at low frequency of N addition (standardized effect size:-0.61, -0.35 and

-0.98 *vs.* -0.58, not significant and -0.60, respectively). Water-soluble S concentration

was directly and negatively affected by both adsorbed and insoluble S at both N

frequencies (Fig. 7a, b).

## 4 Discussion

### 4.1 Positive effect of N addition intensity on soil available S resulted from higher abiotic dissolution, adsorption and organic S mineralization

As expected, increase of water-soluble S was partially due to the dissolution of soil

insoluble S with increasing N addition intensity. Water-soluble S is the most active

and mobile S fraction in topsoil for it can be easily utilized by plants and leached

along with soil pore water (Tabatabai, 2005). Under natural conditions, free sulfate in

soil could also precipitate as calcium-, magnesium- or sodium sulfate and

co-crystallize/co-precipitate with $CaCO_3$ (Tisdale et al., 1993), especially in this

calcareous soil rich in exchangeable Ca, Mg and Na (Wang et al., 2018). However,

under excessive N input, insoluble-S dissolution could sequentially enhance $SO_4^{2-}$

mobility as affected by soil acidification. This postulation was further confirmed by

the significant relationships of soil pH with insoluble S (positive) at low N frequency

and with available S concentrations (negative) at both N frequencies (Fig. 7a,b).

In this study, higher adsorbed S under N addition was mainly derived from higher

ability of sulfate adsorption with decrease of soil pH, which was in line with Nodvin et

al. (Nodvin et al., 1986). Adsorption of $SO_4^{2-}$ is pH dependent as anionic groups from





SOM compete with $SO_4^{2-}$ for adsorption sites on Fe- and Al-hydroxides (Johnson and

Todd, 1983). Under acidic conditions, soil matrix can provide adsorption sites with

positive charges to attract the negatively charged $SO_4^{2-}$ (Tabatabai, 2005). Therefore,

lower soil pH contributed to higher adsorbed S concentration via enhancing adsorption

strength and increasing electrostatic potential of the adsorption sites under higher

intensity of N addition (Scherer et al., 2012).

Nitrogen addition potentially increased organic S mineralization as indicated by

the increased total inorganic S but unchanged (unmown plots) or even decreased

(mown plots) total S concentration. This was consistent with previous observations

where N addition enhanced mineralization of organic S to increase S availability by

elevating microbial activity (Ghani et al., 1992). Soil N availability would also have

considerable impacts on the mineralization of organic S (Gharmakher et al., 2009). The

increases of soil TIN following N input possibly accelerated organic S mineralization

in this study. Moreover, higher plant S uptake under N input (Fig. 6) could promote

biochemical mineralization (McGill and Cole, 1981).

**4.2 Effects of N addition frequency on soil inorganic S fractions**

Partially contrary to our second hypothesis, low frequency of N addition decreased

water-soluble S, available S (only in mown plots) and total inorganic S (mown plots)

concentrations as compared to high frequency (Fig. 2a, b, d; Fig. 5b). Soil available S

concentration was mainly determined by the input from dissolution of insoluble S and

output to plant uptake and leaching. There is a sharper pH decrease with increasing N



intensity under low frequency of N addition, as compared to that under high frequency

of N addition (Ning et al., 2015; Wang et al., 2018). In this study, lower soil pH

decreased insoluble S concentration by promoting its dissolution (Fig. 2e) and

increased S adsorption (Fig. 2d) in unmown plots at low frequency of N addition.

Therefore, lower frequency of N addition stimulated the transformation of insoluble S

into adsorbed S. Another potential explanation could be that large-pulse water input

resulted in higher leaching loss of water-soluble S during N addition at low frequency

than the high N frequency treatment by adding small-amount water each time. Indeed,

infrequent and extreme rainfall pulses have been found to increase $SO_4^{2-}$ leaching in

sandy soil (Eriksen and Askegaard, 2000). Moreover, plant biomass and plant S

uptake were promoted by large dose of N application at low frequency of N addition

(Fig. 6 and S1), which could potentially stimulate the dissolution of insoluble S and

mineralization of organic S (Hu et al., 2002; McGill and Cole, 1981). Therefore,

significantly lower water-soluble S and available S could be a result of higher amount

of S output from plant retention and leaching than input from insoluble S dissolution

under lower frequency of N addition.

   Negative correlation between water-soluble S and adsorbed/ insoluble S

concentrations (Fig. 7b) indicated the complementary roles of these fractions under

high frequency of N addition. However, at lower frequency of N addition, weak

relationships among soil S fractions (including inorganic and organic S, Fig. 7a)

suggested that intense S output pathways broken the coupling of inorganic S fractions.



This was further confirmed by fewer influencing pathways between S fractions in the SEM at low N frequency than the high frequency (Fig. 7c *vs.* Fig. 7d). Under low frequency of N addition, weak relationships among S fractions could be also attributed to unchanged proportion of water-soluble S (Fig. 4a) due to the trade-off

between its input (i.e. dissolution and mineralization) and output (i.e. adsorption and uptake) processes.

**4.3 Mowing effect and its interaction with N addition**

Mowing decreased inorganic S fractions in all treatments, which supported our third hypothesis. Mowing could alter plant community composition (Lü et al., 2012) and

ecosystem nutrient cycling (Koncz et al., 2015). Decreased soil nutrient availability was found under mowing practice in a similar grassland ecosystem resulting from reduced plant residue return (Lü et al., 2012). Without N addition, mowing showed no impact on the rate of inorganic S transformations as suggested by the unchanged proportion of available S *vs.* insoluble S in spite of the decrease in concentrations of

inorganic S fractions (Fig. 4 and 5). Unaffected proportion of inorganic S fractions suggested that biomass removal alone did not stimulate the transformation of S fractions from unavailable forms to available ones. This could probably due to replenishment of inorganic S fractions from organic S mineralization with the presence of relatively low plant S uptake by dominant species without N addition (Fig.

6).

However, when N was added, negative mowing effect on soil available S was



suggested to be exacerbated due to enhanced plant S uptake coincident with higher

plant biomass (Jackson, 2000). This was only the case at low frequency of N addition

in this study, but high N frequency alleviated negative mowing effect and even turned

into positive (> 15 g N m$^{-2}$ yr$^{-1}$) with the increasing N addition intensity (Fig. 3c). The

discrepancy might be recognized as the tradeoff between available S output process

and input process and differential pH responses under two N addition frequencies.

Evidence from SEM also supported this speculation, where direct and negative effects

of mowing on all inorganic S fractions were lower at high frequency of N addition

than the low frequency (Fig. 7). This could be due to higher intensity of plant S

uptake promoted inorganic S transformation and strengthened the relationship

between S fractions at lower frequency of N addition (Fig. 6).

Mowing interacted with N addition increased S uptake by dominant species (Fig.

5) which was probably due to higher S concentration in *Stipa grandis* (Fig. S2) and

increases of biomass proportion of *Stipa grandis* after mowing (Fig. S3). Therefore,

the increase of S removal from the grassland ecosystem could potentially stimulate

available S formation via abiotic dissolution of insoluble S and organic S

mineralization. It was reasonable to detect the decrease of soil total S and insoluble S

but increase of water-soluble S and adsorbed S concentrations under N addition and

mowing treatment.

**5 Conclusions**

Increasing the intensity of N input enhanced soil available S fractions through directly

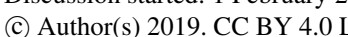



and indirectly affecting soil pH, TIN and insoluble S concentration. Dissolution of

insoluble S and mineralization of organic S contributed to the increases of soil S

availability. Mowing significantly decreased soil inorganic S fractions by reducing S

replenishment via plant residue return and such effect was exacerbated with

increasing intensity of N addition by enhancing plant S uptake. Frequency of N

addition also interacted with mowing to decrease soil adsorbed and available S with

higher response ratio under low frequency of N addition. Our results indicated that

simulating N deposition using large and infrequent pulses of N could overestimate

changes in adsorbed S and available S under unmowing treatment, but underestimate

responses of water-soluble S, adsorbed S and available S concentrations under

mowing treatment. Mowing should be considered as an essential factor in regulating

the effects of N addition intensity and frequency on soil S dynamics in semi-arid

grassland ecosystems. The study could provide insights for sustainable grassland

management in terms of fertilization and mowing practices by concerning their

influences on ecosystem S cycling.

**Acknowledgements**

This work was financially supported by the National Natural Science Foundation of

China (31870441 and 31770525) and the National Key Research and Development

Program of China (2016YFC0500707).

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

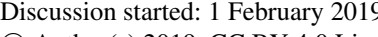



**Table 1** Effects of N addition and mowing practice on soil pH, soil organic carbon
(SOC) and total inorganic nitrogen (TIN). Data shown are the mean values, and the
values in parentheses are standard errors (n = 10), which were previously reported in
Wang et al. (2018).

| | N rate | Unmown | | Mown | |
|---|---|---|---|---|---|
| | (g N m$^{-2}$year$^{-1}$) | F2 | F12 | F2 | F12 |
| pH | 0 | 7.49 (0.13)a | 7.34 (0.08)a | 7.38 (0.12)ab | 7.47 (0.12)ab |
| | 1 | 7.60 (0.18)a | 7.34 (0.08)a | 7.58 (0.13)a | 7.28 (0.11)ab |
| | 2 | 7.40 (0.17)a | 7.51 (0.19)a | 7.61 (0.18)a | 7.64 (0.12)a |
| | 3 | 7.48 (0.17)a | 7.33 (0.14)a | 7.36 (0.21)ab | 7.34 (0.14)ab |
| | 5 | 6.78 (0.17)b | 6.99 (0.18)a | 6.97 (0.18)abc | 7.30 (0.14)ab |
| | 10 | 6.47 (0.19)Bb | 7.08 (0.18)Aa | 6.80(0.16)ABbcd | 7.01 (0.17)Ab |
| | 15 | 6.24 (0.24)Bb | 7.08 (0.18)Aa | 6.42(0.30)ABcd | 6.48 (0.18)ABc |
| | 20 | 5.56 (0.16)c | 6.22 (0.23)b | 6.13 (0.31)de | 6.08 (0.24)c |
| | 50 | 5.21 (0.26)c | 5.13 (0.15)c | 5.54(0.32)e | 5.41(0.29)d |
| TIN | 0 | 9.30 (0.61)c | 10.13 (0.72)d | 9.78 (0.65)d | 9.20 (0.89)c |
| (mg | 1 | 10.29 (1.02)c | 10.24 (1.00)d | 10.13 (0.53)d | 9.74 (0.68)c |
| kg$^{-1}$) | 2 | 11.09 (1.00)c | 9.64 (0.62)d | 10.97 (0.63)d | 10.88 (0.82)c |
| | 3 | 10.19 (1.13)c | 10.61 (0.87)d | 11.94 (0.80)d | 10.75 (0.55)c |
| | 5 | 14.62 (1.34)Ac | 13.19(0.88)ABd | 14.70(1.22)Ad | 10.29(1.00)Bc |
| | 10 | 17.31 (3.38)c | 22.51 (2.47)c | 18.54 (2.92)cd | 17.80 (2.31)bc |
| | 15 | 30.69 (5.46)b | 31.04 (3.77)b | 32.22 (4.41)bc | 25.84 (4.44)b |
| | 20 | 41.03 (5.77)b | 36.48 (4.29)b | 36.46 (7.05)b | 28.05 (3.58)b |
| | 50 | 104.09(10.62)ABa | 74.60 (6.22)Ba | 111.94(15.14)Aa | 71.93(10.32)Ba |
| SOC | 0 | 2.42 (0.06)ABa | 2.17 (0.05)Cb | 2.60 (0.09)Aa | 2.22(0.11)BCab |
| (%) | 1 | 2.45 (0.08)a | 2.24 (0.06)b | 2.45 (0.09)ab | 2.33 (0.15)ab |
| | 2 | 2.17 (0.07)ABb | 2.22 (0.06)ABb | 2.36 (0.11)Aabc | 2.07 (0.08)Bb |
| | 3 | 2.16 (0.07)b | 2.36 (0.13)ab | 2.36 (0.11)abc | 2.27 (0.10)ab |
| | 5 | 2.19 (0.06)Bb | 2.53 (0.09)Aa | 2.27 (0.08)Bbc | 2.35(0.08)ABab |
| | 10 | 2.26 (0.09)ab | 2.39 (0.10)ab | 2.25 (0.08)bc | 2.47 (0.09)a |
| | 15 | 2.12 (0.08)b | 2.21 (0.11)b | 2.14 (0.11)bc | 2.35 (0.10)ab |
| | 20 | 2.43 (0.09)Aa | 2.33(0.08)ABab | 2.10 (0.09)Bc | 2.27(0.11)ABab |
| | 50 | 2.31 (0.05)ab | 2.18 (0.10)b | 2.28 (0.07)bc | 2.36 (0.07)ab |

5 Notes: Different letters indicate significant differences among means of different N addition
frequencies and mowing practice under the same N intensity (uppercase letters) and among means
of different N addition intensities within one frequency of N addition with or without mowing



(lowercase letters). F2 and F12 represent low and high frequency of N addition, respectively.



**Table 2** Results of three-way ANOVAs (*F* value) for the effects of mowing practice (M) and N addition frequency (F) and intensity (N) on soil sulfur (S) fractions and total S concentration.

|        | Water-soluble S | Adsorbed S | Available S | Insoluble S | TIS       | Total S |
|--------|-----------------|------------|-------------|-------------|-----------|---------|
| M      | 74.46**         | 60.40**    | 145.25**    | 231.36**    | 343.35**  | 2.91    |
| F      | 120.82**        | 0.23       | 31.59**     | 1.85        | 25.22**   | 0.08    |
| N      | 3.95**          | 11.36**    | 16.64**     | 12.62**     | 5.94**    | 1.48    |
| M×F    | 4.80*           | 5.98*      | 12.59**     | 5.52*       | 1.31      | 0       |
| M×N    | 1.22            | 0.84       | 0.66        | 3.69**      | 2.62**    | 0.80    |
| F×N    | 5.05**          | 4.83**     | 2.53*       | 0.79        | 2.46*     | 0.39    |
| M×F×N  | 1.45            | 6.65**     | 6.34**      | 1.40        | 1.54      | 1.42    |

Note: TIS represents soil total inorganic S;

5    * and ** represent significance levels ($P \leq 0.05$ and 0.01, respectively)





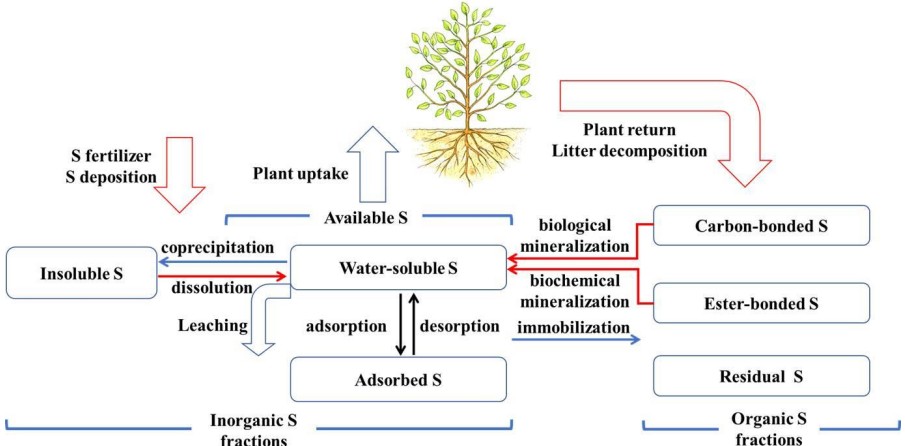

**Figure 1** Conceptual scheme depicting the transformation of sulfur fractions in
aerobic calcareous soils. Arrows indicate input (red) and output (blue) processes of
soil total S (hollow) and available S (solid). Soil available S mainly occur as
water-soluble (the most active fraction) and adsorbed S in aerobic soils. The adsorbed
S is mainly controlled by reversible adsorption-desorption processes which are
pH-dependent. Organic S can be mineralized into inorganic S through biological
process (defined as microbial C oxidation of C-bonded S ) and biochemical process
(extracellular enzymatic hydrolysis of easter-bonded S).




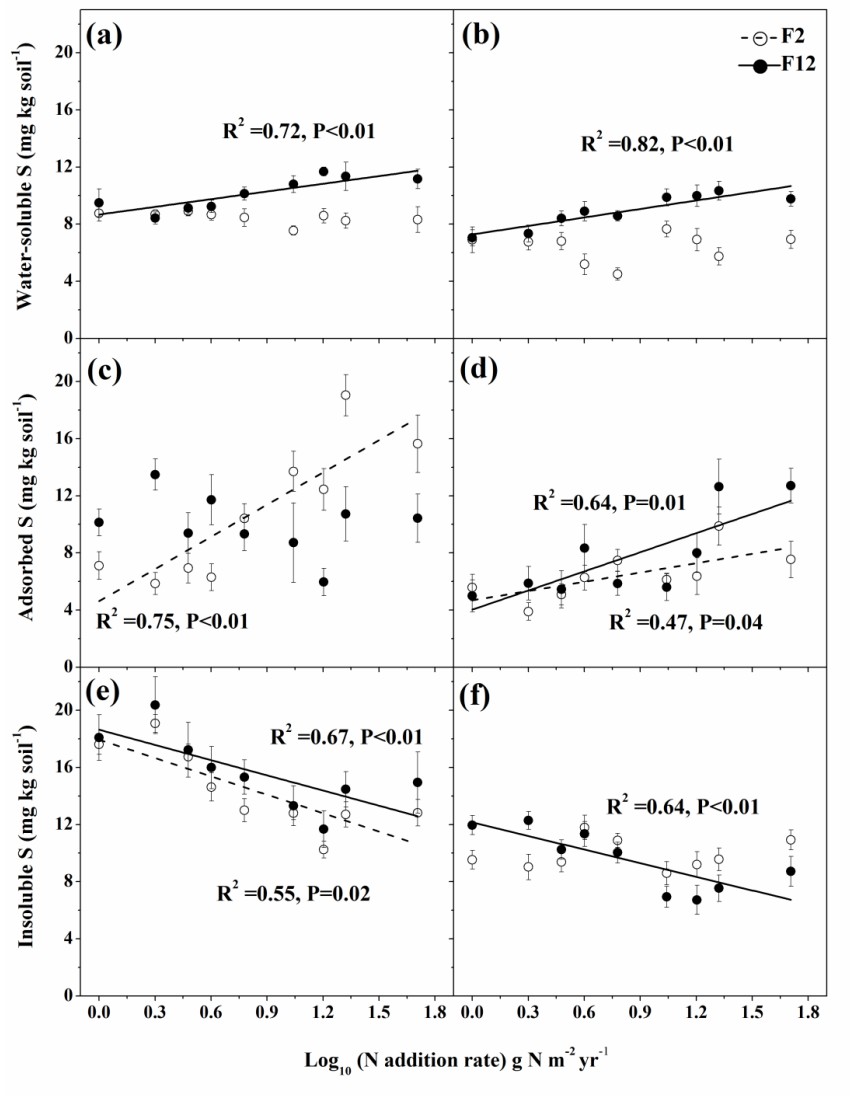

**Figure 2** Effects of N addition intensity and frequency on concentrations of soil
water-soluble S (a, b), adsorbed S (c, d) and insoluble S (e, f) in unmown (left figures)
and mown plots (right figures). Dashed and solid regression lines represent 2 and 12 N

5   additions year$^{-1}$, respectively. Error bars indicate standard error.



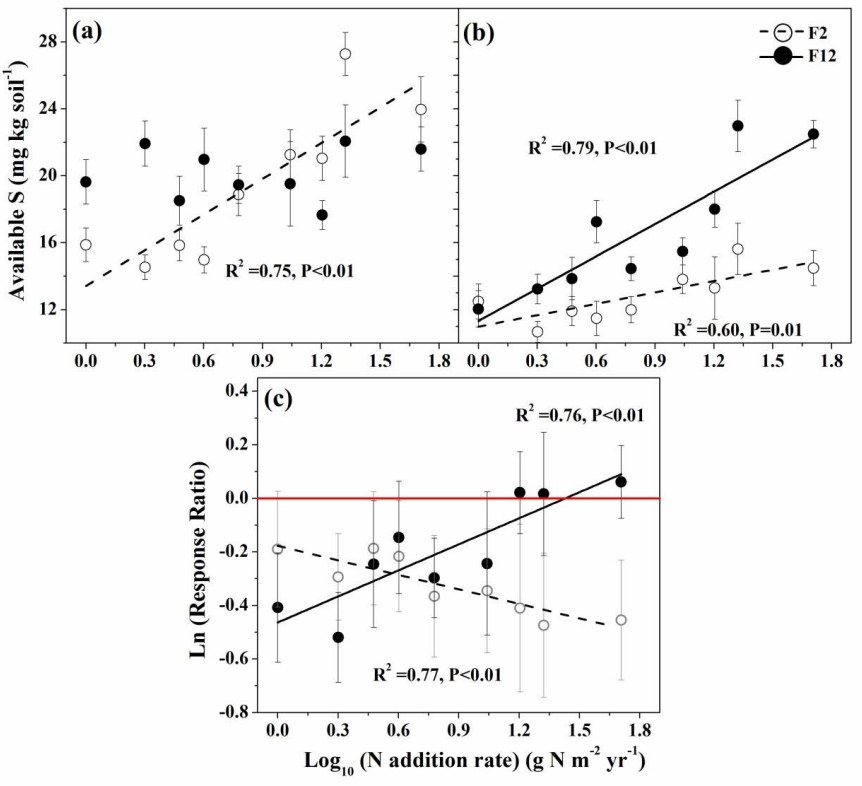

**Figure 3** Effects of N addition intensity and frequency on soil available S concentration

in unmown (a) and mown plots (b) and response ratio of soil available S to mowing

practice along the N addition rate (c). Dashed and solid regression lines represent 2 and

12 N additions year$^{-1}$, respectively. Error bars indicate standard errors in figures (a) and

(b), and 95% confidence intervals in figure (c).

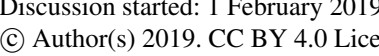



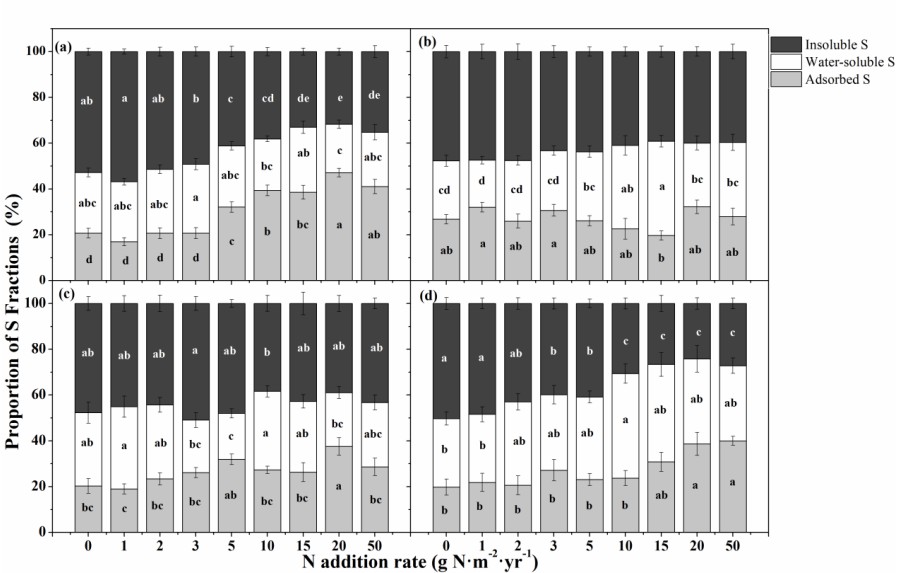

**Figure 4** Proportion of inorganic sulfur fractions relative to total inorganic sulfur under

low and high frequency of nitrogen (N) addition in unmown (a and b, respectively) and

mown (c and d, respectively) plots. White, gray and black bars correspond to

5  water-soluble S, adsorbed S and insoluble S, respectively. Error bars indicate standard

errors. Different letters represent significant difference among N addition rates for each

inorganic S fraction.



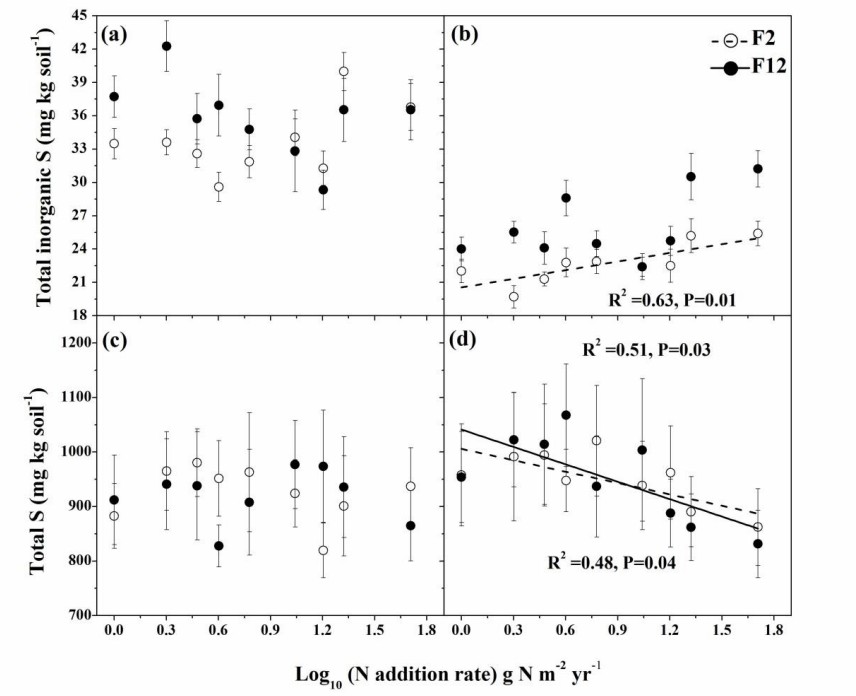

**Figure 5** Effects of N addition intensity and frequency on concentrations of soil total

inorganic S (a, b) and total S (c, d) in unmown (left) and mown (right) plots. Dashed and

solid regression lines correspond to 2 and 12 N additions year$^{-1}$, respectively. Error bars

5    indicate standard errors.





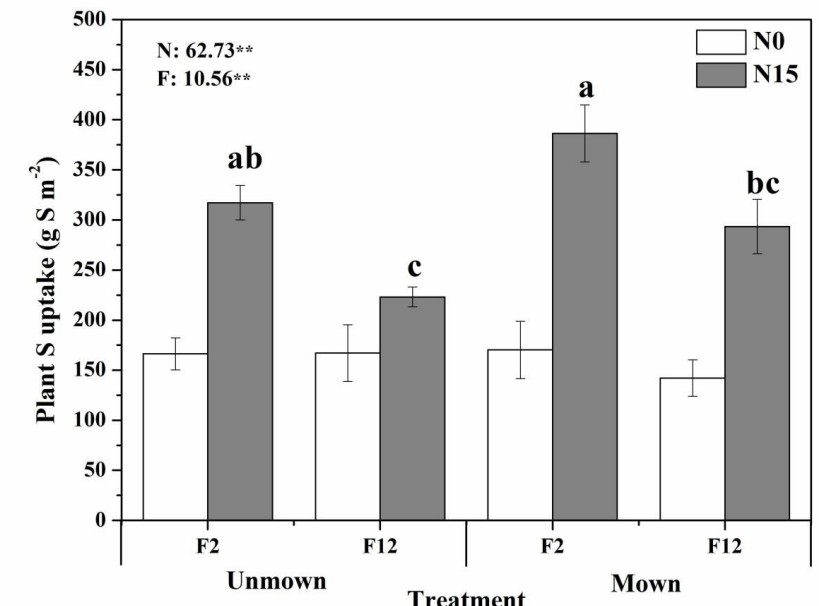

**Figure 6** Effect of N addition intensity and frequency on plant S uptake by dominant

species of *Leymus chinensis* and *Stipa grandis* without and with mowing (only plant

samples in 0 and 15 g N m$^{-2}$ year$^{-1}$ were measured). F2 and F12 indicate low and high

5    frequency of N addition, respectively. Error bars indicate standard error. * and **

represent significance levels ($P < 0.05$ and 0.01, respectively). Different letters above

the bars represent significant difference among means for frequency of N addition and

mowing treatments within 0 or 15 g N m$^{-2}$ year$^{-1}$.



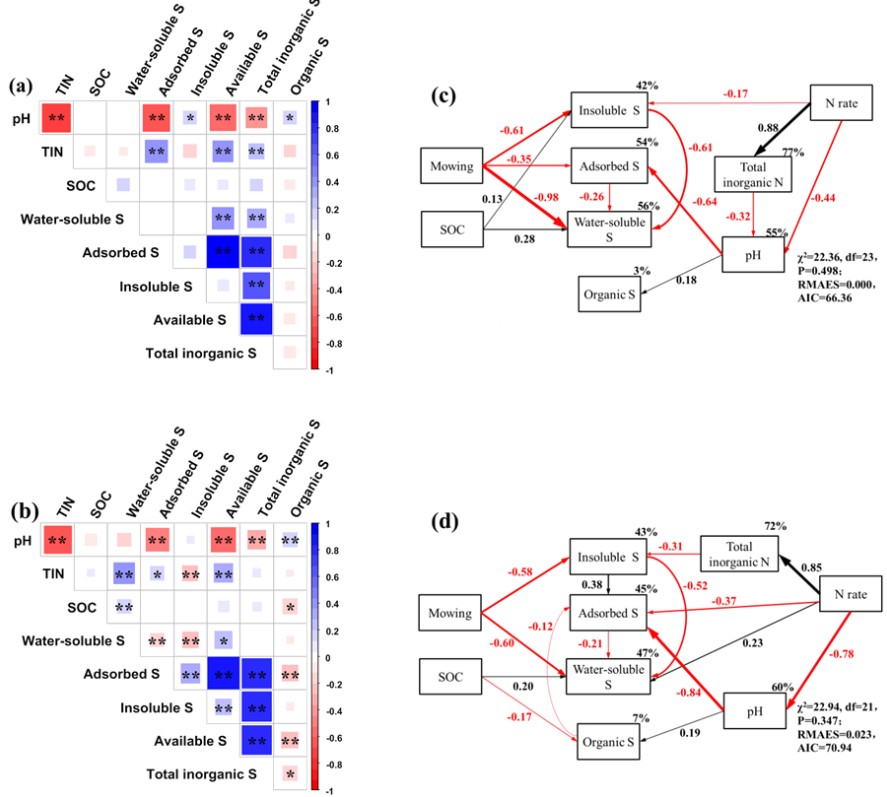

**Figure 7** Correlations between soil parameters and inorganic S fractions with low (a) and high (b) frequency of N addition across N addition intensity and mowing treatments; and structural equation modelling (SEM) illustrating the pathways of effects of N addition intensity and mowing on soil parameters and inorganic S fractions under low (c) and high (d) frequency of N addition. For the correlation, * and ** represent significance levels of $P < 0.05$ and 0.01, respectively. For the SEM, arrows indicate significantly positive (black) and negative (red) effects with the effect size proportional to arrow width. Numbers adjacent to arrows were standardized path coefficients, and percentages close to endogenous variables indicated the variance explained by the model ($R^2$). The final SEM fitted the data well as suggested by the fitting parameters in the figure.