# Peer review of "Frequency and intensity of nitrogen addition alter soil inorganic sulfur fractions but the effects vary with mowing management in a temperate steppe"

_Biogeosciences, 2018_

## Referee Comment (RC1) · Anonymous Referee #1 · 26 Feb 2019

General comment: This paper investigated the effect of N loads on S dynamics in grassland soil and shows interesting topics. However, there are some concerns especially for data analysis. For example, the authors defined total inorganic S as HCl extractable S + $PO_4$ extractable S. But, these fractions would highly overlap each other (See below comments). Therefore, the authors should re-analyze and revise related data and discussion. After the revision, further review on discussions is needed.

Specific comments: P.2 L11-13: This sentence is misleading because the concentrations of water-soluble S and adsorbed S did not always increase with N intensity. This

sentence would be integrated with L17-20 and described accurately.

P.5 L2: Bobbink et al., 2010 is not listed in reference section.

P.6 L9: Is mowing really common in temperate grasslands of the world? You should add a reference to support this sentence.

P.8 L10-11: The experimental design is interesting. But, readers needs to be provided with far more information on the experimental design, how to decide the intensities and frequencies of N addition and what do these correspond to?

P.8 L21: Is this sand S free?

P.10 L7: Please add a reference.

P.10 L15, 17, 20 21: Mistypes. Please put a space before a unit.

P.10 L20: What is the concentrations of H2O2? Does this method include extracted organic S into inorganic S?

P10 L21: What is acacia solution?

P11 L3-4: According to Roberts and Bettany, 1985, which you cited, total inorganic S was defined as HCl-extractable S and insoluble inorganic S was HCl- extractable S – water-soluble S. HCl solution can extract both water-soluble S and adsorbed S by dissolving clay minerals. It would be better that total inorganic S was defined as HCl-extractable S and insoluble inorganic S was HCl-extractable S – available S.

P.11 L9: Please add a reference.

P.11 L15: Although you did several statistical analyses, have you been checked by an expert on statistical analysis? I'm not sure these analyses are right. However, at least for the SEM model, it would be necessary to describe which data set is used for, what the initial model is and how to select the paths from the model. Is Duncan's HSD (I'm sorry but I don't know this method. Is it same to Duncan's multiple range test?)

available for proportional data?

P.12 L14: When using R, you should add a reference of "R core team".

P.18 L19 - P.19 L6: Do you have an idea why adsorbed S concentration in unmown plots at low frequency didn't increase with changes in soil pH?

P.19 L14-15: Unclear.

P.19 L19-21: How to conclude this? Soil available S is affected by various factors such as mineralization rate, plant uptake, amount of adsorption material and soil chemistry. More detailed explanation is needed.

P.20 L6-8: When discussing the effect of experimental operation, you should use the results in control plots.

P.20 L14; Compared for what?

P21 L14: Does this proportion reflect the S transformation? What is the definition of transformation rate in your study? This proportion may be affected by various factors such as mineralization rate, plant uptake, amount of adsorption material and soil chemistry. More detailed explanation is needed.

Figure 6: It is difficult to understand which pair is compered and which letter corresponds.

Figure 7: This figure too small to see. The asterisk in blue column is hard to see.

---

## Referee Comment (RC2) · Anonymous Referee #2 · 3 Apr 2019

Dear authors,

I reviewed this manuscript entitled "Frequency and intensity of nitrogen addition alter soil inorganic sulfur fractions but the effects vary with mowing management in a temperate steppe" by Li et al. submitted to Biogeoscicences as a discussion paper. This study assessed the responses of soil sulfur (S) dynamics to mowing, intensity of nitrogen (N) load, and frequency of N addition. The theme is interesting and data obtained from long-term extensive field experiment are valuable and I think it is suitable for the readership of Biogeoscicences. However, the manuscript was quite complicating because there were various form of S and many combinations of treatments; numbers of N addition levels, two types of frequencies of fertilizer addition (F2 or F12), and mowing or un-mowing. Unfortunately, I think most of the readers of Biogeosciences are familiar with either N dynamics or S dynamics, so the authors should take more cares to induce readers to understand your manuscript more smoothly. Here, I provide some comments which I hope you to improve your manuscript.

Major comments

1. To support your view and/or hypotheses of S dynamics and interactions between many forms of S, Fig 1 should be more highlighted in Introduction and Methods sections, and should be involved with procedures of extraction and calculation of S forms; I think it is necessary to discriminate what form of S was analyzed directly by extraction procedure and what form of S was calculated indirectly from concentrations of analyzed forms.

2. The path structure of SEM analysis and underlying idea should be introduced in Methods section (P12 L15~). The variables can be divided into three categories [related to practices (mowing, N rate), independent variables (pH, TIN, SOC etc.), and independent ones (forms of S)], while all of the items are boxed in same way in the current figures (Fig. 7c, d). Please explain the assumptions and/or typical, expected interactions among these items as a status of pre-analysis. It will be also effective to integrate with research hypotheses (in P7 L8~15).

3. Are the treatments of mowing, intensity and frequency of N addition is comparable to the conventional management of the grassland in this region? How much is the amount of N added to the experimental plots compared to N deposition rate in this region and N fertilizers conventionally used for this grasslands?

Specific comments

P10 L8: "nitrite" is NO2-. Here, this may be "nitrate (NO3-)".

P10 L20: What is "acacia solution"? Is this a kind of chemical used for stabilizing solutes?

P11 L3-4: Equations should be enumerated; one equation by one line, and numbered.

P11 L12: What is "i" in this equation? This equation should also be numbered continuously following the previous equations.

P13 L17: Fig. 1b -> Fig. 2b

P14 L8, L16: Are these percentages (55%, 43% and 40%) average among all N addition intensities?

P16 L19: characters -> characteristics

P17 L16-18: I could not understand the indirect positive effect of N rate on adsorbed S from Fig. 7c. Is it mediated by pH? Is "positive" effect derived from two negative effects, N rate -> pH and pH -> adsorbed S? From that interpretation, the direct and indirect effects of N rate on adsorbed S is strange (Fig. 7d); the indirect effects of N rate on adsorbed S mediated by pH should be positive because both arrows are negative, while the direct effects of N rate on adsorbed S is negative.

Fig. 1: It is unclear that Available S is sum of Water-soluble S and Adsorbed S. Also, I could not see the difference between hollow and solid arrows.

Fig. 4: Alphabets indicating significant difference according to multiple comparison should be added to Insoluble S in Fig. 4b.

Fig. 7c, d: "N rate" should be "N addition intensity". Please indicate that the bars right side of Fig. 7a, b, changing color red to blue, represent correlations.

---

## Author Response (AR1)

Dear Professor Nobuhito Ohte,

We thank you very much for considering our manuscript (bg-2018-526), "Frequency and intensity of nitrogen addition alter soil inorganic sulfur fractions but the effects vary with mowing management in a temperate steppe". We appreciate the reviewers for their positive and constructive comments and suggestions on our manuscript.

According to the reviewers' comments, we recalculated the insoluble sulfur (S), total inorganic S and organic S concentration and re-analyzed the data. All the results and discussion were updated accordingly, although the main conclusions of our manuscript did not change. We also revised the conceptual model and structural equation modeling according to the suggestions from both reviewers. We hope the manuscript now meets the level of Biogeosciences. In this revised version, changes to our manuscript within the document were all highlighted by using blue-colored text. Point-by-point responses to the reviewers are listed below, the original comments are in black, and our responses are in blue.

**Reviewer #1:**

**General comment:** This paper investigated the effect of N loads on S dynamics in grassland soil and shows interesting topics. However, there are some concerns especially for data analysis. For example, the authors defined total inorganic S as HCl extractable S + $PO_4$ extractable S. But, these fractions would highly overlap each other (See below comments). Therefore, the authors should re-analyze and revise related data and discussion. After the revision, further review on discussions is needed.

Response: We agree with the comment and feel sorry for the confusion caused by our way of defining soil inorganic S fractions. Following your suggestion, we clarified the definition of each inorganic S fraction and re-analyzed the data. They were clarified as "Water-soluble S, available S and total inorganic S were extracted with 0.01 *M* $CaCl_2$, 0.01 *M* $Ca(H_2PO_4)_2$ and 1 *M* HCl at a 5:1 (w/v) water: soil ratio, respectively (Roberts and Bettany, 1985) (P.11 L11-14)". We listed the equations for calculating

adsorbed S, total inorganic S and organic S concentrations (P.12 L1-5). All the related statistical analyses, including tables and figures and the discussion were updated.

**Specific comments:**

1. P.2 L11-13: This sentence is misleading because the concentrations of water-soluble S and adsorbed S did not always increase with N intensity. This sentence would be integrated with L17-20 and described accurately

Response: Thanks for the suggestion. We corrected the description by integrating the two sentences into 'Generally, N addition frequency, N intensity and mowing significantly interacted with each other to affect most of inorganic S fractions. Specifically, the significant increase of water-soluble S was only found at high N frequency with the increasing intensity of N addition. Increasing N addition intensity enhanced adsorbed S and available S concentrations at low N frequency in unmown plots; however, both fractions significantly increased with N intensity at both N frequencies in mown plots' (P.2 L11-17).

2. P.5 L2: Bobbink et al., 2010 is not listed in reference section.

Response: Added in the reference section. Please see P.26 L17.

3. P.6 L9: Is mowing really common in temperate grasslands of the world? You should add a reference to support this sentence.

Response: Yes, mowing is common in temperate grasslands. It is one of the oldest and most widespread practices in grassland management to produce hay, which can be stored for on-farm/agricultural use. As suggested, references were added (P.6 L16).

*Bremer, D. J., and Ham, J. M.: Measurement and modeling of soil CO2 flux in a temperate grassland under mowed and burned regimes. Ecol. Appl., 12, 1318-1328, https://doi.org/10.1890/1051-0761(2002)012[1318:MAMOSC]2.0.CO;2, 2002.*

*Zhang, Y., Loreau, M., He, N., Zhang, G., and Han, X.: Mowing exacerbates the loss of ecosystem stability under nitrogen enrichment in a temperate grassland, Funct. Ecol., 31, 1637-1646, https://doi.org/10.1111/1365-2435.12850, 2017.*

4. P.8 L10-11: The experimental design is interesting. But, readers need to be provided with far more information on the experimental design, how to decide the intensities and frequencies of N addition and what do these correspond to?

Response: More information on the experimental design was provided (P.8 L20-P.9 L1-6). Higher frequency of N addition is to simulate natural N deposition and to determine whether the effect of frequent N addition differs from infrequent N addition as common used to mimic N deposition in manipulative experiments. Higher rates of N addition were used to mimic accumulative N deposition in the long-term and/or extreme N inputs in the future.

5. P.8 L21: Is this sand S free?

Response: The sand used in this experiment is sulfur free and the information was added (P.9 L18-20).

6. P.10 L7: Please add a reference.

Response: We added a reference for soil organic carbon measurement (P.11 L2).

7. P.10 L15, 17, 20 21: Mistypes. Please put a space before a unit.

Response: These mistypes have been corrected (P.11 L13, L15, L17 L20).

8. P.10 L20: What is the concentrations of $H_2O_2$? Does this method include extracted organic S into inorganic S?

Response: The concentration of $H_2O_2$ is 30% (P.11 L15). This method does not include extracted organic S into inorganic S because the presence of $Ca^{2+}$ in the extractants depresses the solubility of organic matter (P.11 L15-17) including organic S. Therefore, the $BaCl_2$-turbidimetry only determines sulfate ion ($SO_4^{2-}$) extracted from inorganic S fractions.

9. P10 L21: What is acacia solution?

Response: Sorry, it is gum acacia solution, which helps stabilize the suspension (P.11

L18).

10. P11 L3-4: According to Roberts and Bettany, 1985, which you cited, total inorganic S was defined as HCl-extractable S and insoluble inorganic S was HCl-extractable S − water-soluble S. HCl solution can extract both water-soluble S and adsorbed S by dissolving clay minerals. It would be better that total inorganic S was defined as HCl-extractable S and insoluble inorganic S was HCl-extractable S − available S.

Response: Thanks for the helpful suggestion. We agree with the point that HCl can extract both water-soluble S and adsorbed S. As suggested, we recalculated the data by defining total inorganic S as HCl-extractable S (P.11 L11-12) and insoluble S as HCl-extractable S − available S (P.12 L3). We updated all the related statistics including the tables and figures accordingly.

11. P.11 L9: Please add a reference.

Response: Added as suggested (P.12 L9).

12. P.11 L15: Although you did several statistical analyses, have you been checked by an expert on statistical analysis? I'm not sure these analyses are right. However, at least for the SEM model, it would be necessary to describe which data set is used for, what the initial model is and how to select the paths from the model. Is Duncan's HSD (I'm sorry but I don't know this method. Is it same to Duncan's multiple range test?) available for proportional data?

Response: We carefully checked our statistical analyses and consulted an expert on statistical analysis to make sure these are right. More information about the SEM model were provided: 1) the dataset that we used for running the SEM model were described in the main text (P.13 L19); 2) we added the information of what the initial model is and how to select the paths from the model (P.13 L20-P.14 L4 and caption of Fig S1). We employed Duncan's multiple range test (P.13 L5) instead of Duncan's HSD (sorry for the typo) for analyzing proportional data.

13. P.12 L14: When using R, you should add a reference of "R core team".

Response: As suggested, the reference of "R core team" were provided in P.13 L14.

14. P.18 L19 - P.19 L6: Do you have an idea why adsorbed S concentration in unmown plots at high frequency didn't increase with changes in soil pH?

Response: At high N frequency, unaffected adsorbed S in unmown plots could be possibly due to the fact that soil pH tended to be higher as compared to low N frequency at the same N addition level (statistically significant at 10, 15 and 20 g N $m^{-2}$ $yr^{-1}$, see Table 1). Moreover, soil pH decreased at a much lower rate along with increasing N addition intensity (significant decrease only detected at 20 and 50 g N $m^{-2}$ $yr^{-1}$) under high N frequency comparing to low N frequency. This resulted in weaker S adsorption strength, less $SO_4^{2-}$ release from insoluble S dissolution and consequently non-significant increase of adsorbed S concentration with increasing N intensity at high N frequency. This explanation has been added (P.19 L18-P.20 L5).

15. P.19 L14-15: Unclear.

Response: As suggested, we clarified the statement into "*Moreover, decrease of soil pH and higher plant S uptake under N input (Fig. 5) could promote biochemical mineralization of organic S via enhancing secretion of arylsulfatase by soil microorganisms (McGill and Cole, 1981). This further confirmed with Niknahad et al. (2009) reporting the upregulation of soil organic S mineralization by decrease of soil pH*" (P.21 L13-18).

16. P.19 L19-21: How to conclude this? Soil available S is affected by various factors such as mineralization rate, plant uptake, amount of adsorption material and soil chemistry. More detailed explanation is needed.

Response: Thanks for the helpful suggestion. As per suggestion, we provided detailed explanation by relating to factors of soil chemistry (soil pH decline and insoluble S dissolution), mineralization, leaching and plant S uptake (P.22 L8-P.23 L1). The

explanation was written as "*A sharper decrease of soil pH with low N frequency relative to high N frequency was expected to result in higher soil available S concentration via enhanced insoluble S dissolution. In contrary, lower available S concentration was found under low N frequency as compared to high N frequency, which could be possibly driven by higher plant S uptake (Fig. 5) surpassing the amount of S dissolution. Leaching loss of $SO_4^{2-}$ was suggested to be evident with infrequent and extreme rainfall pulses in sandy soils (Eriksen and Askegaard, 2000). Therefore, another potential explanation could be that large-pulse water input at low N frequency resulted in higher leaching loss of available S than the high N frequency of adding small-amount water each time. And the results of adsorbed S, available S and total inorganic S in the control plots supported this explanation. With the increase of N intensity, leaching loss of available S was exacerbated due to the fact of enhancing insoluble S dissolution (Fig. 2e,f). Comparing to high N frequency, organic S mineralization did not contribute to lower total inorganic S and available S concentrations at low N frequency for the same N intensity as no difference was detected for organic S concentration between two N frequencies*".*

And we concluded this as "*These results suggest that using low frequency of N addition to mimic N deposition may overestimate the processes of insoluble S dissolution (especially in unmown plots) and plant S uptake in temperate grasslands. However, some of the S fractions responded differently to both N intensity and frequency with or without mowing treatment suggesting that the effects of N addition strongly depended on mowing practice*" (P.23 L1- 6).

17. P.20 L6-8: When discussing the effect of experimental operation, you should use the results in control plots.

Response: As suggested, we mentioned the results in control plots when discussing the effect of experimental operation (P.22 L16-17). Further, we found leaching loss of available S was exacerbated due to the fact of enhancing insoluble dissolution (P.22 L17-19).

18. P.20 L14; Compared for what?

Response: We compared the treatment of low N frequency with high N frequency. Similar comparisons were clarified throughout this section (P.22 L4-P.23 L6).

19. P21 L14: Does this proportion reflect the S transformation? What is the definition of transformation rate in your study? This proportion may be affected by various factors such as mineralization rate, plant uptake, amount of adsorption material and soil chemistry. More detailed explanation is needed.

Response: We sincerely appreciate the valuable comments. After carefully considering this issue, we think it is not appropriate to define transformation rate as changes in the proportion of S fractions. Therefore, we re-plotted our data and changed the explanation into "*mowing resulted in significant decreases of inorganic S fractions and the proportion of total inorganic S relative to soil total S. In contrast, relative organic S proportion increased with mowing treatment, which indicates that mowing management removes soil S out of the ecosystem having a larger impact on inorganic S transformation rather than the organic S mineralization*" (P.23 L12-17).

20. Figure 6: It is difficult to understand which pair is compered and which letter corresponds.

Response: We feel sorry for the confusion. We have clarified it into "Different letters above the bars represent significant difference among means for the N addition frequency (F2 *vs*. F12) at 15 g N m$^{-2}$ yr$^{-1}$ (N15) with and without mowing separately. No significant difference was detected between the two N frequencies at 0 g N m$^{-2}$ year$^{-1}$ for both mown and unmown treatments (N0)" (P.42 L7-11).

21. Figure 7: This figure too small to see. The asterisk in blue column is hard to see.

Response: We divided the correlation heat map and SEM into two separated Figures (Figure 6 and Figure 7). We colored the asterisk in yellow to ensure it visible in both red and blue columns.

**Reviewer #2:**

**Major comments:**

1. To support your view and/or hypotheses of S dynamics and interactions between many forms of S, Fig 1 should be more highlighted in Introduction and Methods sections, and should be involved with procedures of extraction and calculation of S forms; I think it is necessary to discriminate what form of S was analyzed directly by extraction procedure and what form of S was calculated indirectly from concentrations of analyzed forms.

Response: Thanks for the helpful suggestion. As per suggestion, we incorporated related information in Figure 1 to discriminate which form of S was analyzed directly by extraction procedure and which S was calculated indirectly from concentrations of analyzed forms. We highlighted Figure 1 in both Introduction (P.4 L16-20) and methods sections (P.11 L10).

2. The path structure of SEM analysis and underlying idea should be introduced in Methods section (P12 L15~). The variables can be divided into three categories [related to practices (mowing, N rate), independent variables (pH, TIN, SOC etc.), and dependent ones (forms of S)], while all of the items are boxed in same way in the current figures (Fig. 7c, d). Please explain the assumptions and/or typical, expected interactions among these items as a status of pre-analysis. It will be also effective to integrate with research hypotheses (in P7 L8~15).

Response: As suggested, we introduced the path structure of SEM analysis and underlying idea by building a *priori* model in Method section (Fig. S1a; P.13 L20-P.14 L4). Here, we combined the three categories of variables in one model (Fig. S1a) and added the expected interactions among these items and described as "soil S fractions could be directly affected by N addition frequency, intensity and mowing, or indirectly by altering soil pH, plant biomass return and organic S mineralization" (P.13 L20-P.14 L2). Moreover, we integrated these expected interactions with our hypotheses as described in the caption of Fig. S1. To obtain the best-fit final model,

insignificant pathways and parameters that had no effect on inorganic fractions were excluded from the model sequentially (see Fig. S1b, c). This information was also mentioned in P.14 L2-4.

3. Are the treatments of mowing, intensity and frequency of N addition is comparable to the conventional management of the grassland in this region? How much is the amount of N added to the experimental plots compared to N deposition rate in this region and N fertilizers conventionally used for this grassland?

Response: We sincerely appreciate the valuable comment. Mowing for hay harvesting by local people in late August is very common in this region (P.6 L15-17). Nitrogen deposition rate is about $1\sim2$ g N$^{-2}$ year$^{-1}$ in this area which equal to the low N addition intensity of this experiment. We added higher amount of N to mimic accumulative N deposition in the long-term and/or extreme N inputs in the future (P.9 L5-6). Due to the fact that infrequent N addition (i.e. 1 or 2 time per year) is commonly used in manipulative experiments to mimic N deposition, a more frequent and even way of N addition (i.e. 12 times yr$^{-1}$, high N frequency) was set to simulate natural N deposition to compare whether changing frequency of N input would affect grassland ecosystem. This information has been added in subsection of 'experimental design' (P.8 L20-P.9 L4).

**Specific comments:**

1. P10 L8: "nitrite" is $NO_2^-$. Here, this may be "nitrate ($NO_3^-$)".

Response: Thanks. We have corrected the "nitrite" into "nitrate" (P.11 L4).

2. P10 L20: What is "acacia solution"? Is this a kind of chemical used for stabilizing solutes?

Response: Yes, the gum acacia solution was used to stabilize the solutes. This information has been added in P.11 L18.

3. P11 L3-4: Equations should be enumerated; one equation by one line, and

numbered.

Response: As per suggestion, each equation has been numbered in a separated line (P. 12 L2-4 & L13).

4. P11 L12: What is "i" in this equation? This equation should also be numbered continuously following the previous equations.

Response: "$i$" denotes the plant species $i$, which has been defined in P.12 L14. All the equations have been numbered continuously now.

5. P13 L17: Fig. 1b -> Fig. 2b

Response: We corrected "Fig. 1b" into "Fig. 2b" (P.15 L3).

6. P14 L8, L16: Are these percentages (55%, 43% and 40%) average among all N addition intensities?

Response: We calculated the percentages within each N intensity and N frequency. 55% (now it's 95% after data re-analysis following the suggestion from Reviewer #1) is the highest percentage change among all N treatments across both N intensity and frequency. 43% and 40% are the highest percentage changes among N intensities for low N frequency and high N frequency, respectively. These have been clarified in the main text (P.16 L2-3).

8. P16 L19: characters -> characteristics

Response: We changed "characters" into "characteristics" (P.14 L9; P.18 L3).

9. P17 L16-18: I could not understand the indirect positive effect of N rate on adsorbed S from Fig. 7c. Is it mediated by pH? Is "positive" effect derived from two negative effects, N rate -> pH and pH -> adsorbed S? From that interpretation, the direct and indirect effects of N rate on adsorbed S is strange (Fig. 7d); the indirect effects of N rate on adsorbed S mediated by pH should be positive because both arrows are negative, while the direct effects of N rate on adsorbed S is negative.

Response: Thanks for mentioning this. For the indirect effect, two negative effects indeed result in one positive effect but it's still an indirect effect; and the total effect size depends on the relative size of direct and indirect effects. After carefully considering this comment and the general comment #3, we re-ran the SEM model by combining three treatments, independent soil variables and dependent ones and then corrected our interpretation (P.18 L15-P.19 L4).

10. Fig. 1: It is unclear that Available S is sum of Water-soluble S and Adsorbed S. Also, I could not see the difference between hollow and solid arrows.

Response: The Figure 1 has been modified by involving with procedures of extraction and calculation of S forms. Related information has been added in Methods section (P.11 L9-13 and P.11 L21-P.12 L4) and in caption of Figure 1. We utilized green and red arrows to represent opposite processes affecting soil S fractions.

11. Fig. 4: Alphabets indicating significant difference according to multiple comparison should be added to Insoluble S in Fig. 4b

Response: In the previous version, we did not label with alphabets where insoluble S concentrations were insignificant among N intensities. After considering the comments from Reviewer #1, we recalculated the proportional data as inorganic S fractions relative to total S concentration because proportions of S fractions could not reflect their transformation very well. The figure has been moved to supplementary material as Figure S5 and all significant difference has been labeled.

12. Fig. 7c, d: "N rate" should be "N addition intensity". Please indicate that the bars right side of Fig. 7a, b, changing color red to blue, represent correlations

Response: As suggested, "N rate" has been corrected into "N addition intensity". We added the description of the changing color red to blue of the bars representing correlations.

With above corrections, the manuscript is hereby resubmitted to the journal. We are thankful for the reviewers' work and glad to respond any further questions that you have. We look forward a positive response from you.

Thanking you,

**Ruzhen Wang**

ruzhenwang@iae.ac.cn

Institute of Applied Ecology, Chinese Academy of Sciences

---

## Referee Report (RR1)

I have only checked the responses to my review comments (Reviewers 1). The authors have done a good job in responding to my queries and requests but I have some additional queries.

1. P.11 L10: Soluble S or labile S may be better than water-soluble S, because $CaCl_2$ is not just water.

2. P.11 L15-17: Although $Ca^{2+}$ can depress the concentration of organic matter, it could not completely remove organic matter. That's why you used $H_2O_2$ to remove organic matter, right? And how is HCl extraction? it does not include $Ca^{2+}$.
   Although below is just suggestion, if you have next opportunity of the experiment, it is better to apply another method, for example using a resin like DAX-8.

3. P.22 L10-11: Please add corresponding No. of figure or table. Figure 3?

4. P.22 L6: Available S concentration itself is mainly determined by soil characteristics especially amount of adsorption material. Perhaps what you want to say is a change in concentration?

5. P.22 L16-17: How does the result of control plot support your hypothesis? The explanation is insufficient.